# Occurrence, Pathogenic Potential and Antimicrobial Resistance of *Escherichia coli* Isolated from Raw Milk Cheese Commercialized in Banat Region, Romania

**DOI:** 10.3390/antibiotics11060721

**Published:** 2022-05-27

**Authors:** Kálmán Imre, Alexandra Ban-Cucerzan, Viorel Herman, Khalid Ibrahim Sallam, Romeo Teodor Cristina, Samir Mohammed Abd-Elghany, Doru Morar, Sebastian Alexandru Popa, Mirela Imre, Adriana Morar

**Affiliations:** 1Faculty of Veterinary Medicine, Banat’s University of Agricultural Sciences and Veterinary Medicine “King Michael I of Romania” Timişoara, 300645 Timișoara, Romania; kalmanimre@usab-tm.ro (K.I.); alexandra_cucerzan@yahoo.com (A.B.-C.); viorel.herman@fmvt.ro (V.H.); romeocristina@usab-tm.ro (R.T.C.); doru.morar@yahoo.com (D.M.); popa_sebastian_alexandru@yahoo.com (S.A.P.); mirela.imre@gmail.com (M.I.); 2Faculty of Veterinary Medicine, Mansoura University, Mansoura 35511, Egypt; khalidsallam@mans.edu.eg (K.I.S.); drsamir@mans.edu.eg (S.M.A.-E.)

**Keywords:** virulence, antimicrobial resistance, cheese, Romania, *Escherichia coli*

## Abstract

The aim of the present study was to investigate the presence, pathogenic potential and antimicrobial susceptibility profile of *Escherichia coli* isolated from raw milk cheese, traditionally produced by farmers and marketed directly to the consumer in Banat region, Romania. A total of 81.1% (43/53) of the processed samples expressed positive results for *E. coli*, with a distribution of 83.8% (31/37), and 75.0% (12/16) in the cow- and sheep-milk-origin assortments, respectively. Overall, 69.8% (30/43) of the specimens had a contamination level ≤10 CFU/g. Molecular tests showed that, from the total number of *E. coli* isolates, 9.3% (4/43) harbored the *stx*2, and 2.3% (1/43), the *stx*1 virulence genes. The *E. coli* O157 (including H7) biovariety was identified in 7% (3/43) of the samples by the Vidas equipment. From the 27 antimicrobials tested with the Vitek2 automated system, the *E. coli* isolates displayed resistance to enrofloxacin (100%, 15 out of 15 tested isolates), ampicillin (39.5%, 17/43), norfloxacin (28.6%, 8/28), fosfomycin (25%, 7/28), amoxicillin/clavulanic acid (23.3%, 10/43), cefalexin (20%, 3/15), cefalotin (13.3%, 2/15), tetracycline (13.3%, 2/15), trimethoprim-sulfamethoxazole (9.3%, 4/43), piperacillin-tazobactam (7.1%, 2/28), cefotaxime (7.1%, 2/28), cefepime (7.1%, 2/28), ticarcillin/clavulanic acid (6.7%, 1/15), florfenicol (6.7%, 1/15), ceftazidime (3.6%, 1/28), and ertapenem (3.6%, 1/28). Ten (23.3%) strains were multidrug-resistant. The obtained preliminary results indicated hygienic–sanitary deficiencies throughout the cheese production process, and demonstrated that these products can harbor virulent and multidrug-resistant *E. coli* strains, which constitute a public health risk. However, future investigations, processing a higher number of samples, are still necessary to draw comprehensive conclusions.

## 1. Introduction

Raw-milk-origin dairy products, and especially cheeses, can be contaminated with a large variety of bacteria that are potentially harmful to humans, including the enteric pathogen *Escherichia coli*. This rod-shaped, Gram negative and facultatively anaerobic microorganism is considered a normal inhabitant of the large intestinal tract of warm-blooded animals. It is eliminated in the external environment, together with feces, which can result in the contamination of raw milk, as a consequence of improper hygiene practices during the milking process [1,2].

The results of several epidemiological investigations reviewed by Kaper et al. [3] have demonstrated the link between the occurrence of severe food-borne disease outbreaks, or even deaths, and the consumption of raw dairy products contaminated with different diarrheagenic *E. coli* strains. Studies focused mainly on monitoring the Shiga-toxin-producing *E. coli* (STEC) strains found in food matrices, due to their toxicity and infectious potential, related to the toxin productions (e.g., Stx1 or Stx2), which generate hemorrhagic colitis, as well as two other major lethal syndromes in human patients, namely, thrombotic thrombocytopenic purpura and hemolytic uremic syndrome [4,5,6].

The traditional production of raw milk cheese in small-scale integrated livestock farms from rural areas, and their marketing within agri-food markets to the urban consumers, is a widely spread practice in several countries, including Romania [7]. However, it is possible for the resulted products to harbor a variety of harmful microorganisms, including *E. coli* [8,9,10,11,12].

In recent decades, the inappropriate use of antimicrobial agents for different purposes (e.g., therapeutic, prophylactic), has led to the appearance of an undesired consequence: the so-called antimicrobial resistance (AMR) phenomenon, in the case of both commensals and bacteria that threaten public health. In Romania, a recent review highlighted a distressing AMR profile of the major foodborne pathogenic bacteria [13], strengthening the necessity for continuous surveillance of the diversity and dispersion of drug-resistant bacteria within a tripartite intersectoral chain (human–animal–environment), especially regarding finished products. At present, no scientific report is available in our country on the phenotypic resistance pattern of the cheese-origin *E. coli* strains. Therefore, the present study was undertaken to investigate the occurrence, pathogenic potential, and antimicrobial susceptibility profile of *E. coli* isolated from raw milk cheese produced and marketed in the Banat region, Romania.

## 2. Results

Microbiological analysis of 53 raw milk cheese specimens showed that 43 (81.1%; 95% CI 68.6–89.4) were contaminated with *E. coli*, with a distribution of 31 (83.8%; 95% CI 68.9–92.4), and 12 (75.0%; 95% CI 50.5–89.8) positive samples for the cow and sheep milk origin assortments, respectively. These differences are not statistically significant (*p* > 0.05). The results regarding the *E. coli* strains are reported in Table 1. Overall, 30 (69.8%; 95% CI 54.9–81.4) of the raw milk cheeses harbored ≤10 CFU/g of *E. coli*, while a significantly lower (*p* < 0.05) number of samples (*n* = 13) exceeded this contamination level. Furthermore, the recorded results showed a relatively uniform *E. coli* contamination level of the tested samples according to the milk origin (cow or sheep) (Table 1).

Molecular confirmation of the isolated and biochemically confirmed *E. coli* strains, based on the PCR amplification of the partial sequence of the 16S rRNA gene, was successfully achieved for all (*n* = 43) isolates. Monitorisation of the presence of two major virulence markers lead to the detection of the *stx*2 gene in 9.7% (3/31) and 8.3% (1/12) of *E. coli* isolates recovered from cow- and sheep-milk-origin cheese samples, respectively. In addition, one (3.2%) cow-milk-origin isolate harbored the *stx*1 gene. 

The presence of the *E. coli* O157 (including H7) biovariety was confirmed in 7% (3/43) of the screened samples (Table 1) with the help of the Vidas system, based on cutting-edge phage recombinant protein technology, as well as the use of the recommended chromogenic medium.

Results from the antimicrobial resistance profile monitoring of the isolated *E. coli* strains (*n* = 43), according to the specific bacteria cards used in this study, are summarized in Table 2 and Table 3. For isolates (*n* = 28) processed with the AST-N204 card, resistance was noticed against AMP (42.8%), NOR (28.6%), AMC (25.0%), FOF (25%), SXT (10.7%), TZP (7.1%), CTX (7.1%), CMP (7.1%), ERT (3.6%), and CAZ (3.6%), but none of the isolates were resistant against AMK, GEN, MEM, IPM, CIP, and NIT, respectively. Furthermore, the rest of the isolates (*n* = 15), tested with the AST-GN96 card, were resistant towards ENR (100%), AMP (33.3%), AMC (20.0%), LEX (20%), CET (13.3%), TET (13.3%), TIM (6.7%), FFC (6.7%), and SXT (6.7%), but all of the isolates were susceptible to GEN, TIO, CEF, MBX, and UB. Likewise, notable susceptibility rates were recorded for IPM (93.3%) and NEO (93.3%).

Unfortunately, ten (23.3%; 95% CI 13.2–37.8) of the tested *E. coli* isolates (8 from cow and 2 from sheep milk) were classified as MDR because they revealed resistance to at least one antimicrobial agent from three or more classes. Each of them expressed different resistance profiles (Table 4).

## 3. Discussion

In agreement with the results of several previous investigations [9,14,15], our study pointed out that the isolation of *E. coli* strains from raw milk cheese samples is a common finding in the surveyed region (Banat) from Romania. The overall 81.1% isolation frequency rate that was recorded suggested that most of these assortments were produced in improper hygiene and sanitary conditions. This contamination rate is higher than previously reported rates for raw milk telemea cheese (another white brined traditional product) in Romania (62.5%; [9]), or for white cheese in Turkey (60%; [16]), but lower than rates found in Minas Frescal (100%; [17]), and soft cheese assortments (97.7%; [18]) in Brazil. Furthermore, the study showed that 68.9% of positive samples had an *E. coli* contamination level below 10 CFU/g. As term of comparison, Brooks et al. [14] reported that 95% (39/41) of the examined bovine, caprine or ovine origin raw milk cheese specimens in the U.S., had *E. coli* counts < 10 CFU/g. Similar findings were obtained by O’Brien et al. [19], who observed that 79% of the investigated raw milk cheeses in Ireland had *E. coli* counts < 10 CFU/g. It is important to mention that *E. coli* infection in humans requires low infectious doses, estimated at <100 cells [4]. However, the reported results are markedly influenced by the study design, seasonal and geographic variations, the testing methods (including the culture media and growth temperatures) or the differences in hygiene and milking practices [20].

A striking finding was yielded during molecular screening of the virulence pattern of isolated *E. coli* strains. Thus, 5 (11.6%) out of 43 isolates harbored one of the targeted genes encoding toxin secretion. These results highlight that the investigated raw milk cheese samples may constitute a potential public health risk [21]. Similarly, the occurrence of *stx*-positive raw-milk-/cheese-origin *E. coli* strains was previously confirmed in several studies, with very different detection rates in Egypt (1/140-0.7% [8]), Iran (11/11-100% [11]), India (25/80-31.3% [22]), and in another study conducted in Romania (27/145-18.6% [9]). Even if the presented preliminary results of the present study can be considered suggestive, further investigations focusing on the molecular evidence of other enterohemorrhagic (e.g., *eaeA*), enterotoxigenic (e.g., *elt*, *est*), enteroinvasive (e.g., *invE*), enteroaggregative (e.g., *Eagg*, *astA*), and diffusely adherent (e.g., *daaD*) virulence factors/genes are still required for a better understanding of the pathogenic potential of raw-milk-cheese-origin *E. coli* strains. Furthermore, the positive findings (7%) obtained in terms of *E. coli* O157 (including H7) detection via the Vidas system and subsequent confirmation of the specific recommended chromogenic medium, constitute further proof that the monitored cheese products can be a reservoir for virulent enterohemorrhagic *E. coli* strains. Similar findings have also been pointed out by other investigations [23,24].

The expressed antimicrobial pattern of the tested *E. coli* strains (*n* = 43) indicated a wide variability of resistance against 16 (59.3%) of a total of 27 tested antimicrobial substances. Four antimicrobials, included in both utilized specific cards, with different mic range values, accounted for the highest resistance rates, namely: AMP (39.5%), followed by AMC (23.3%) and SXT (9.3%), but no resistance was recorded against GEN. In agreement with these findings, different resistance levels for the raw-milk/cheese-origin strains have been recorded for AMP in Ukraine (100%, [25]), China (46.3%, [26]), and India (8.9%, [22]); for AMC in Iran (100%, [11]), and Brazil (40.0% [27]); for SXT in Turkey (44.2%, [16]), and China (13.4%, [26]); and for GEN in Iran (100%, [10]), Turkey (53.7%, [16]), and India (3.3%, [22]). The recorded resistance for AMP, AMC and SXT in the present survey can be explained by the unrestricted and preferred usage of these drugs in the management of mammary gland infections by veterinarians in rural farming in Romania (Imre, unpublished results). Nonetheless, further research, enrolling a higher number of samples, should be carried out to obtain a truthful conclusion. 

In addition, the tested *E. coli* strains exhibited different levels of resistance towards six out of eight cephalosporins, namely, LEX (20.0%), CET (13.3%), CTX (7.1%), CPM (7.1%), and CAZ (3.6%). Therefore, it is worth noting that these results constitute a great concern for public health, because cephalosporins (3rd and 4th generations) are widely used in the management of human infections at present. Furthermore, the resistance levels reported in the present study against ENR (100%), and remarkable in the case of NOR (28.6%), FOF (25.0%), and TET (13.3%) are quite disturbing. Different resistance rates have been observed for these drugs in several previous studies [10,11,25,26,27].

A considerable number of isolated *E. coli* strains (23.3%) expressed MDR, presenting different resistance profiles towards antimicrobials belonging to six common classes of antimicrobials (Table 2 and Table 3) used for evaluation. The occurrence of the MDR *E. coli* strains is not surprising, considering that largely similar results have been reported in several previous studies. Thus, the recorded MDR resistance rate was higher in Iran (30.2%, [10]) and Saudi Arabia (100%, [28]), but lower in Northern China (19.4%, [26]). However, to obtain a better understanding of the AMR phenomenon of this food-borne pathogen in our country, further studies, based on the monitoring and characterization of the genotypic resistance pattern of the isolates, by highlighting the presence of antimicrobial resistance genes (e.g., *aadA1, tetA, tetB, dfrA1, qnrA, aaC, bla_TEM_,* or *sul1*), are still necessary. 

In this research, the susceptibility rates recorded for some tested antimicrobials, (e.g., aminoglycosides, carbapenems, quinolones and nitrofuran derivatives classes), can constitute useful tools for physicians and veterinary practitioners, helping to avoid undesired outcomes in terms of *E. coli* infection management.

## 4. Materials and Methods

### 4.1. Sample Collection and Bacterial Isolation

A total of 53 samples, including raw-milk cow (*n* = 37), and sheep (*n* = 16) semisoft unsalted traditional cheese (the so-called *caș*- in Romanian), sold directly, without packaging, from a refrigerator, were randomly collected from nine agri-food markets, situated in different localities from the Banat region located in southwestern Romania. The samples came from a total of 42 producers. Each was sampled during a single visit from March 2020 to April 2021. The investigated cheese specimens were produced by local farmers in small-scale integrated dairy production units no longer than one day prior to the sampling date. The aseptically collected samples (~250 g) were individually packed in a polyethylene bag, labeled with the assortment name and sampling date, and transported under refrigeration conditions (≤4 °C) to the laboratory of Food Hygiene and Microbiological Risk Assessment, Faculty of Veterinary Medicine, Timișoara. Subsequent to their arrival in the laboratory, the samples were subjected to bacterial investigations.

The isolation of *Escherichia coli* was performed according to the steps stipulated by the International Organization and Standardization (ISO) 16649-2/2007, with slight amendments [29]. In brief, ten grams of each cheese sample were homogenized in a Stomacher (bioMérieux, Marcy l’Etoile, France) (120 s) with 90 mL of preheated (45 °C) peptone-buffered solution (PBS; pH = 7.5 ± 0.1). Next, serial dilutions of up to 10^−3^ (1:1000) in sterile 0.5% peptone water were prepared. Subsequently, 1 mL from each dilution was transferred in a sterile Petri dish (in duplicate), and a pre-cooled (44–47 °C) chromogenic selective medium, consisting of tryptone bile agar with X-glucoronide (TBX) agar (~15 mL) (Oxoid, Basingstoke, Hampshire, UK), containing 5-bromo-4-cloro-3-indolil-β-d-glucoronat was poured on top. The resulting mixture was incubated after solidification, first at 37 °C, for 4 h, and then at 43.5 °C, for 24 h, for bacterial isolation. The blue–green-colored presumptive *E. coli* colonies that grew from the 10^−3^ dilution were examined using Gram staining and enumerated according to ISO 16649-2/2007 [29]. Species were identified by testing the complete biochemical properties of the isolates with the Vitek2 (bioMérieux, Marcy l’Etoile, France) automated compact system, along with the reference strain *E. coli* ATCC 25922 that served as a control and using the Vitek 2^®^ ID-GN (Gram-negative, ID Card No. 21311) specific identification cards, following the manufacturer’s instructions. This card includes a total of 47 biochemical tests, in addition to one negative control well measuring the carbon source utilization, enzymatic activities and resistance of the *E. coli* isolates.

### 4.2. Molecular Analyses

One isolate from each *E. coli* positive samples (*n* = 43) underwent molecular testing. For the bacterial genomic DNA isolation, the PureLink^TM^ Genomic DNA Mini Kit (Invitrogen™, Carlsbad, CA, USA) was used, according to the manufacturer’s instructions. Firstly, molecular confirmation of the isolated *E. coli* strains was carried out through a conventional uniplex polymerase chain reaction (PCR), based on the amplification of a partial sequence of the 16S rRNA gene, and using the universal oligonucleotide primer sets and cycling conditions, according to the methodology reported by Magray et al. [30]. Next, the presence of two major virulence markers produced by enterohemorrhagic *E. coli* strains, namely, *stx*1 (Shiga toxin 1), and *stx*2 (Shiga toxin 2) typical toxin genes, were screened using the specific primer set and cycling conditions designed by Blanco et al. [31]. The used primer sequences and PCR conditions are available in the Appendix A. Genomic DNA obtained from *E. coli* O157 ATCC35401, harboring both virulence genes, was used in all reactions as a positive control, and PCR-grade water (no template DNA), as a negative control. In all cases, the amplified PCR products were analyzed via electrophoresis in 1.8% agarose gel, stained with Midori Green (Nippon Genetics^®^; Europe, Gmbh, Düren, Germany).

### 4.3. Shiga Toxin Detection Immunofluorescence Screening of E. coli O157 (Including H7)

Cheese samples (25 g) found to be positive for *E. coli* were screened for the occurrence of somatic “O157”, as well as flagellar “H7” antigens, using the Vidas^®^ UP *E. coli* O157 (including H7) (ECPT) (bioMérieux, Marcy l’Etoile, France) assay. The tests were performed with the Mini VIDAS (bioMérieux, Marcy l’Etoile, France) automated system, according to the manufacturer’s recommendations. The principle of this qualitative test is based on the Enzyme Linked Fluorescent Assay (ELFA) method. Within this technique, the instrument uses a Solid Phase Receptacle (SPR), coated inside with recombinant phage tail fiber protein to capture the pathogen. Positive or negative results are automatically quantified by the system. Subsequently, all the obtained positive results were confirmed on CHROMID^TM^ EHEC AGAR (bioMérieux, Marcy l’Etoile, France).

### 4.4. Antimicrobial Susceptibility Testing

The antimicrobial susceptibility pattern of the isolated *E. coli* strains was determined using the Vitek2 automated equipment (bioMérieux, Marcy l’Etoile, France). Within this study, in the context of very limited financial resources, the research team tried to expand the diversity of the used antimicrobials to provide feedback to both physicians, as well as veterinarians, on the resistance of the isolated strains towards routinely used antimicrobials. In this regard, two different Gram-negative specific cards, namely, AST-N204 (for human use) and AST-GN96 (for veterinary use), were applied to monitor the antimicrobial resistance profiles of 28 and 15 isolates, respectively. The used cards included a total of 27 antimicrobial substances, from 11 classes, as follows: β-lactams–ampicillin [AMP; tested concentrations: 4, 8 32], amoxicillin/clavulanic acid [AMC; 4/2, 16/8, 32/16], piperacillin-tazobactam [TZP; 2/4, 8/4, 24/4, 32/4, 32/8, 48/8], ticarcillin/clavulanic acid [TIM; 8/2, 32/2, 64/2]; aminoglycosides–amikacin [AMK; 8, 16, 64], gentamicin [GEN; 4, 16, 32], neomycin [NEO; 8, 16, 64]; amphenicols–florfenicol [FFC; 1, 4, 8]; carbapenems–ertapenem [ERT; 0.5, 1, 6], meropenem [MEM; 0.5, 2, 6, 12], imipenem [IPM; 1, 2, 6, 12]; cephalosporins–cefotaxime [CTX; 1, 4, 16, 32], ceftazidime [CAZ; 1, 2, 8, 32], cefepime [CPM; 2, 8, 16, 32]; cefalexin [LEX; 8, 32, 64], cefalotin [CET; 2, 8, 32], ceftiofur [TIO; 1, 2], cefquinome [CEF; 0.5, 1.5, 4]; fluoroquinolones–ciprofloxacin [CIP; 0.5, 2, 4], norfloxacin [NOR; 1, 8, 32], enrofloxacin [ENR; 0.25, 1, 4], marbofloxacin [MBX; 1, 2]; quinolones-flumequine [UB; 2, 4, 8]; nitrofuran derivative–nitrofurantoin [NIT; 16, 32, 64]; phosphonic acid derivative–fosfomycin [FOF; 8, 16, 32]; sulfonamides–trimethoprim-sulfamethoxazole [SXT; 1/19, 4/76, 16/304]; and tetracyclines-tetracycline [TET; 2, 4, 8]. For each drug, the correspondent minimum inhibitory concentration [MIC] range is presented in Table 2 and Table 3. The results for the tested antimicrobials were automatically expressed by the Vitek2 system as susceptible, resistant, and intermediately resistant. *E. coli* isolates showed resistance to at least one antimicrobial agent, in three or more antimicrobial classes, and were classified as multidrug-resistant (MDR) [32]. The resistance breakpoint of the isolates was established in agreement with the European Committee on Antimicrobial Susceptibility Testing (EUCAST) and Clinical Laboratory Standards Institute (CLSI) guidelines [33,34]. The used minimum inhibitory concentration (MIC) breakpoints for the tested antimicrobials, in accordance with the correspondent references, are available in the Appendix A. The *E. coli* ATCC 25922 strain was used as an internal quality control.

### 4.5. Statistical Analysis

Differences in the distribution of *E. coli* strains, according to their sample origin, were statistically analyzed using the Pearson’s chi-square (χ^2^) test. The results were considered statistically significant for *p* ≤ 0.05.

## 5. Conclusions

The preliminary results of the present survey emphasize a high contamination rate with *E. coli* of raw milk cheese, traditionally produced in the Banat region, Romania, indicating hygienic–sanitary deficiencies during the milking procedure and/or cheese production, handling or selling. Likewise, the evidence of potentially virulent genes in some *E. coli* strains could lead to a public health hazard, considering that the monitored cheese products can constitute transmission vehicles for pathogenic bacteria. Furthermore, the occurrence of MDR strains represents an important public health issue, highlighting that continuous efforts are needed (e.g., the careful use of antimicrobials or respecting the principles of antibiotic prescription) to restrict the development of the AMR phenomenon. The study results also call for the attention of veterinary inspectors and require them to be more stringent when monitoring the effectiveness of the hygiene measures implemented by farmhouse cheesemakers during production and marketing. However, future investigations, processing a higher number of samples, are still necessary to draw comprehensive conclusions.

## Figures and Tables

**Table 1 antibiotics-11-00721-t001:** Isolation and contamination level with *E. coli* of raw milk cheese assortments, and detection of virulence genes carrying isolates using molecular and immunological tools.

Origin of Cheese Samples (No. of Examined)	No. of Samples Containing *E. coli* (%)	No. and (%) of Samples with Different Levels of *E. coli* (CFU/g)			No. of *E. coli* Isolates Carrying Virulence Genes (%)		No. of Positive Samples for *E. coli* O157:H7 (%)
		≤10	>10 ≤100	>100	*stx*1	*stx*2	
cow milk (*n* = 37)	31 (83.8)	24 (77.4)	5 (16.1)	2 (6.4)	1 (3.2)	3 (9.7)	2 (6.5)
sheep milk (*n* = 16)	12 (75.0)	6 (50.0)	2 (16.7)	4 (33.3)	-	1 (8.3)	1 (8.3)
Overall (*n* = 53)	43 (81.1)	30 (69.8)	7 (16.3)	6 (13.9)	1 (2.3)	4 (9.3)	3 (7.0)

Legend: No—number of samples; CFU—colony forming unit.

**Table 2 antibiotics-11-00721-t002:** Antimicrobial susceptibility profile of the Romanian fresh-cheese-origin *E. coli* strains tested with the AST-N204 card.

Antimicrobial			Susceptibility Test Result of 28 Strains (*n*/%)	
Class	Agent	MIC Range µL/mL	S	R
β-lactams	AMP	≤2	16 (57.2)	12 (42.8)
	AMC	≤2	21 (75.0)	7 (25.0)
	TZP	≤4	26 (92.9)	2 (7.1)
aminoglycosides	AMK	≤2	28 (100)	-
	GEN	≤1	28 (100)	-
carbapenems	ERT	≤0.5	27 (96.4)	1 (3.6)
	MEM	≤0.25	28 (100)	-
	IPM	≤0.25	28 (100)	-
cephalosporins	CTX	≤1	26 (92.9)	2 (7.1)
	CAZ	≤1	27 (96.4)	1 (3.6)
	CPM	≤1	26 (92.9)	2 (7.1)
fluoroquinolones	CIP	≤0.25	28 (100)	-
	NOR	≤0.5	20 (71.4)	8 (28.6)
nitrofuran derivative	NIT	64	28 (100)	-
phosphonic acid derivative	FOF	≤16	21 (75)	7 (25.0)
sulfonamides	SXT	≤20	25 (89.3)	3 (10.7)

Key: *n* = number of strains expressed susceptibility (S) and resistance (R).

**Table 3 antibiotics-11-00721-t003:** Antimicrobial susceptibility profile of the Romanian fresh-cheese-origin *E. coli* strains tested with the AST-GN96 card.

Antimicrobial			Susceptibility Test Result of 15 Strains (*n*/%)		
Class	Agent	MIC Range µL/mL	S	R	I
β-lactams	AMP	16	10 (66.6)	5 (33.3)	-
	AMC	≥32	12 (80.0)	3 (20.0)	-
	TIM	≤8	13 (86.6)	1 (6.7)	1 (6.7)
aminoglycosides	GEN	≤1	15 (100)	-	-
	NEO	≤2	14 (93.3)	-	1 (6.7)
amphenicols	FFC	4	6 (40.0)	1 (6.7)	8 (53.3)
carbapenems	IPM	4	14 (93.3)	-	1 (6.7)
cephalosporins	LEX	≥64	12 (80.0)	3 (20.0)	-
	CET	≥64	12 (80.0)	2 (13.3)	1 (6.7)
	TIO	≤1	15 (100)	-	-
	CEF	≤0.5	15 (100)	-	-
fluoroquinolones	ENR	≤0.12	-	15 (100)	-
	MBX	≤0.5	15 (100)	-	-
quinolones	UB	≤1	15 (100)	-	-
sulfonamides	SXT	≤20	14 (93.3)	1 (6.7)	-
tetracyclines	TET	4	13 (86.6)	2 (13.3)	-

Key: *n* = number of strains expressing susceptibility (S), resistance (R) or intermediate resistance (I).

**Table 4 antibiotics-11-00721-t004:** Multi-drug resistance phenotype combination of the tested *E. coli* isolates.

Origin of Cheese	No. of Isolates	Vitek Card Used	No. of Classes with Resistance	Resistance to Antimicrobial Profile	Classes with Resistance
Cow milk	1	AST-GN96	4	AMP, AMC, CTX, CAZ, CPM, FOF, SXT	β-lactams, cephalosporins, phosphonic acid derivative, sulfonamides
	1	AST-GN96	4	AMP, AMC, CTX, CPM, FOF, SXT	β-lactams, cephalosporins, phosphonic acid derivative, sulfonamides
	1	AST-GN96	4	AMP, AMC, TZP, FOF, NOR, SXT	β-lactams, phosphonic acid derivative, fluoroquinolones, sulfonamides
	1	AST-N204	4	AMP, ENR, TET, SXT	β-lactams, fluoroquinolones, sulfonamides, tetracyclines
	1	AST-GN96	3	AMP, AMC, TZP, FOF, NOR	β-lactams, fluoroquinolones, phosphonic acid derivative
	1	AST-GN96	3	AMP, FOF, NOR	β-lactams, fluoroquinolones, phosphonic acid derivative
	1	AST-N204	3	AMP, LEX, ENR	β-lactams, cephalosporins, fluoroquinolones
	1	AST-N204	3	AMP, AMC, ENR, TET	β-lactams, fluoroquinolones, tetracyclines
Sheep milk	1	AST-N204	3	AMP, AMC, LEX, CET, ENR	β-lactams, cephalosporins, fluoroquinolones
	1	AST-N204	3	AMP, AMC, TIM, LEX, CET, ENR,	β-lactams, cephalosporins, fluoroquinolones

## Data Availability

All data generated or analyzed during this study are included in the manuscript.

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
