# Peer review of "Occurrence, Pathogenic Potential and Antimicrobial Resistance of Escherichia coli Isolated from Raw Milk Cheese Commercialized in Banat Region, Romania"

_antibiotics, 2022, doi:10.3390/antibiotics11060721_

Round 1
Reviewer 1 Report
In the study, the authors investigated the presence, pathogenic potential and antimicrobial susceptibility profile of Escherichia coli isolates in raw milk cheese, in Romania.
I found this work interesting because it highlights possible public health hazards and hygienic-sanitary failures during cheese production in the area considered for the study.
I have only a consideration about the Antimicrobial suscetibility testing: I think that the authors should explain, in the “Materials and Methods” section, why some strains were analyzed with the Vitek card AST-N204 and other strains with the Vitek card AST-GN96.
Author Response
Dear Editor and Reviewer,
We would like to thank the Editor and each reviewer for their time, interest and comments to our manuscript. We are delighted to be considered for publication in the prestigious Antibiotics-Basel journal and to receive quality peer reviews that have greatly improved our manuscript. We have thoroughly considered each Reviewer's comments and have revised the text to reflect this. We have addressed each Reviewer’s comments below, in a point-by-point fashion. The new version of the manuscript contains all of the requested and operated changes and additional information (as reviewers required) highlighted with red font. Also, following the recommendations of a reviewer, the manuscript was checked by a native English-speaking colleague.
Below, you can find the answers to the all comments and suggestions.
Thank you again for your time and consideration.
Sincerely yours,
Dr. Adriana MORAR
On behalf of all the authors
Reviewer #1
In the study, the authors investigated the presence, pathogenic potential and antimicrobial susceptibility profile of Escherichia coli isolates in raw milk cheese, in Romania.
I found this work interesting because it highlights possible public health hazards and hygienic-sanitary failures during cheese production in the area considered for the study.
I have only a consideration about the Antimicrobial susceptibility testing: I think that the authors should explain, in the “Materials and Methods” section, why some strains were analyzed with the Vitek card AST-N204 and other strains with the Vitek card AST-GN96.
Answer: Thank you very much for your positive appreciations! As the authors highlighted within the lines 281-283 of the original submission, the most of the isolates (n=28) has been tested with the card AST-N204, designed for antimicrobials used in human medicine. However, in order to (i) increase the study value, (ii) to widen the used antimicrobials diversity and (iii), also, to provide feedback to veterinarians, the authors decided that a smaller part of the isolates (n=15) to be tested with a Gram-negative card (AST-GN96) recommended for veterinary use only. The authors acknowledged that it would have been better that all isolates had been tested with both cards, but the existent financial resources were very limited. Precisely for this reason, the study was presented under the form of Communication, presenting preliminary, but significant results. The authors intend to continue the study. According to the reviewer request, the authors included a justification for using different Vitek cards, for testing antimicrobial susceptibility profile of the isolates, namely:
“Within this study, in context of the existence of very limited financial resources, the research team tried to expand the diversity of the used antimicrobials in order to provide feedback to both physicians, as well as veterinarians on the resistance of the isolated strains towards routinely used antimicrobials.”
In this regard, two different Gram-negative specific cards, namely AST-N204 (for human use) and AST-GN96 (for veterinary use), were applied for the monitoring of antimicrobial resistance profile of 28 and 15 isolates, respectively”
Thank you again for your time and efforts!
Reviewer 2 Report
Introduction
Please shorten. A lot of the points discussed are well-known facts, hence no need to burden readers.
Procedures
Table 4. Please transfer to supplementary material.
4.4. Why using the CLSI guidelines? This is a European study, hence the EUCAST guidelines must be employed. Please re-evaluate all the results according to European standards.
Results
Antimicrobial resistance profile of E. coli: please include in a new table.
MDR: please include the official reference for this definition.
Discussion
This is verbose for a rather limited study, hence, it should be restructured by deleting passage not directly relevant to the study performed.
Overall. The manuscript is a limited study: only 53 cheese samples from only one region of the country. As such, it does not justify a full article.
The authors must shorten the manuscript significantly and resubmit as communication.
Moreover, English language is not good, there are many mistakes in the manus. Improvement is paramount.
Author Response
Dear Editor and Reviewer,
We would like to thank the Editor and reviewer for their time, interest and comments to our manuscript. We are delighted to be considered for publication in the prestigious Antibiotics-Basel journal and to receive quality peer reviews that have greatly improved our manuscript. We have thoroughly considered each Reviewer's comments and have revised the text to reflect this. We have addressed each Reviewer’s comments below, in a point-by-point fashion. The new version of the manuscript contains all of the requested and operated changes and additional information (as reviewers required) highlighted with red font. Also, following the recommendations of a reviewer, the manuscript was checked by a native English-speaking colleague.
Below, you can find the answers to the all comments and suggestions.
Thank you again for your time and consideration.
Sincerely yours,
Dr. Adriana MORAR
On behalf of all the authors
Reviewer #2
Introduction
Please shorten. A lot of the points discussed are well-known facts, hence no need to burden readers.
Answer: According to the reviewer requirement the introduction chapter has been significantly shortened. Please see the revised version.
Procedures
Table 4. Please transfer to supplementary material.
Answer: According to the reviewer requirement, the Table 4 was transferred to supplementary material.
4.4. Why using the CLSI guidelines? This is a European study, hence the EUCAST guidelines must be employed. Please re-evaluate all the results according to European standards.
Answer: Thank you for this raised concern! The research team regrettable omitted to specify that, in principle, the antimicrobial susceptibility results were interpreted according to the European Committee on Antimicrobial Susceptibility Testing (EUCAST) guideline. The CLSI guideline was used, only for antibiotics that do not have defined MIC breakpoint correspondent in the EUCAST guideline. This concern was clarified in the revised version of the manuscript, and for a more comprehensive understanding, the used minimum inhibitory concentration (MIC) breakpoints for the tested antimicrobials, in accordance with the correspondent references, were included in Supplementary Materials.
Results
Antimicrobial resistance profile of E. coli: please include in a new table.
Answer: According to the reviewer requirement, the Table 1 has been divided into two tables, resulting a more understandable form of the antimicrobial resistance profile of the E. coli isolates. (Please see Tables 2 and 3 in the revised version). Another reviewer also has this request/suggestion.
MDR: please include the official reference for this definition.
Answer: the official reference for this definition was included in the materials and methods section as “Magiorakos, A.P.; Srinivasan, A.; Carey, R.B.; Carmeli, Y.; Falagas, M.E.; Giske, C.G.; Harbarth, S.; Hindler, J.F.; Kahlmeter, G.; Olsson-Liljequist, B.; et al. Multidrug-resistant, extensively drug-resistant and pandrug-resistant bacteria: An international expert proposal for interim standard definitions for acquired resistance. Clin. Microbiol. Infect. 2012, 8, 268–281. https://doi.org/10.1111/j.1469-0691.2011.03570.x”
Discussion
This is verbose for a rather limited study; hence, it should be restructured by deleting passage not directly relevant to the study performed.
Answer: According to the manuscript preparation guideline of the Antibiotics journal (https://www.mdpi.com/journal/antibiotics/instructions) research manuscript should comprise: Introduction, Results, Discussion, Materials and Methods, Conclusions (optional) sections. During manuscript preparation, the authors tried to fulfill this request. Thank you for your undersatnding!
Overall. The manuscript is a limited study: only 53 cheese samples from only one region of the country. As such, it does not justify a full article. The authors must shorten the manuscript significantly and resubmit as communication.
Answer: The authors want to highlight the fact that during initial submission the manuscript was submitted as Communication (please see the heather – Communication – indicating the type of the manuscript.) In the Instructions for authors section of Antibiotics journal (https://www.mdpi.com/journal/antibiotics/instructions), the following request is defined: “Articles should have a main text of around 3000 words at minimum and should have more than 30 references. Antibiotics has no restrictions on the maximum length of research manuscripts, provided that the text is concise and comprehensive.” Accordingly, the total word count of the initial submission was 3134 (except the abstract, tables and references), with 33 references. The manuscript length was conceived, before its submission, to fulfill this request. Thank you for your understanding!
Moreover, English language is not good, there are many mistakes in the manus. Improvement is paramount.
Answer: According to the reviewer requirement, was checked by a native English-speaking colleague. The resulted modifications have been highlighted with blue font in the revised version of the manuscript. Please verify!
Thank you again for your recommendations, efforts and time!
Reviewer 3 Report
This short communication focused on the isolation and characterization of Escherichia coli from raw milk cheese commercialized in Banat region, Romania. Although this topic is interesting, there is a lack of in-depth experiments regarding pathogenic potential or antimicrobial resistance. Moreover, only a limited number (53) of samples were used to isolate Escherichia coli, which would make the conclusions a little speculative. Therefore, this manuscript is not acceptable in the current form. Detailed comments are as below:
(1) Line 17: The number of samples and strains was not enough to draw a comprehensive conclusion.
(2) Line 20: The molecular studies were relatively simple. As a short communication, you can focus on either virulence potential or antibiotic resistance of Escherichia coli. The pathogenic potential can be determined by virulence gene detection and cell invasion test. In terms of antimicrobial resistance, detection of resistance genes by PCR or WGS is also required.
(3) Line 21: What about the presence of other virulence genes?
(4) Line 74: What was the novelty of this work?
(5) Lines 88-89: What was the minimum number of pathogenic Escherichia coli to cause infections?
(6) Line 125: Was the antibiotic resistance of these isolates more serious than that in similar studies?
Author Response
Dear Editor and Reviewer,
We would like to thank the Editor and reviewer for their time, interest and comments to our manuscript. We are delighted to be considered for publication in the prestigious Antibiotics-Basel journal and to receive quality peer reviews that have greatly improved our manuscript. We have thoroughly considered each Reviewer's comments and have revised the text to reflect this. We have addressed each Reviewer’s comments below, in a point-by-point fashion. The new version of the manuscript contains all of the requested and operated changes and additional information (as reviewers required) highlighted with red font. Also, following the recommendations of a reviewer, the manuscript was checked by a native English-speaking colleague.
Below, you can find the answers to the all comments and suggestions.
Thank you again for your time and consideration.
Sincerely yours,
Dr. Adriana MORAR
On behalf of all the authors
Reviewer #3
This short communication focused on the isolation and characterization of Escherichia coli from raw milk cheese commercialized in Banat region, Romania. Although this topic is interesting, there is a lack of in-depth experiments regarding pathogenic potential or antimicrobial resistance. Moreover, only a limited number (53) of samples were used to isolate Escherichia coli, which would make the conclusions a little speculative. Therefore, this manuscript is not acceptable in the current form.
Answer: Thank you very much for your overall positive appreciations and efforts to review our manuscript!
(1) Line 17: The number of samples and strains was not enough to draw a comprehensive conclusion.
Answer: The research team members completely agree the reviewer opinion that, overall, the number of collected and processed cheese samples is low. This concern, is related by the existent of very limited financial resources. However, the total number of collected samples (n=53) were provided by a representative number of cheese producers from the screened region, offering a relatively complex image in term of territorial distribution about the hygienic – sanitary quality of the cheese assortments which they produce. Also, this fact can result in a more complex understanding of the phenotypic antimicrobial resistance phenomenon of the isolated E. coli strains, originating from cheeses produced by different farmhouse cheesemakers (the herds of different cheese producers receive different antimicrobial treatment protocols). Being a preliminary investigation, the sampling strategy tried to focuse on the increasing of the sample origin diversity, and not on to collect a high number of samples in different periods, within a large interval, from a limited number of producers. In the nearest future, if new financial resources will be identified, the authors intend to continue the begun investigations. In addition, the authors want to highlight the fact that before submission the manuscript quality was presented to the Editor, and after a preliminary checking, and during the authors assimilation process to the western-word, the manuscript submission was encouraged! However, according to the reviewer recommendation, some sentences were reformulated, in terms of „preliminary results”, highlighting the fact that future investigations, processing a highest number of samples, are still necessary to draw a valid comprehensive conclusion.
(2) Line 20: The molecular studies were relatively simple. As a short communication, you can focus on either virulence potential or antibiotic resistance of Escherichia coli. The pathogenic potential can be determined by virulence gene detection and cell invasion test. In terms of antimicrobial resistance, detection of resistance genes by PCR or WGS is also required. (3) Line 21: What about the presence of other virulence genes?
Answer: The authors acknowledge the fact that the study presents limited molecular biology and antimicrobial susceptibility monitoring methodologies, somewhat offset by immunofluorescence techniques. Part of these limitations has been highlighted in the original submitted version of the manuscript. „However, for a better understanding of the AMR phenomenon of this food-borne pathogen in our country, further studies, based on the monitoring and characterization of genotypic resistance pattern of the isolates, through evidentiating antimicrobial resistance genes (e.g. aadA1, tetA, tetB, dfrA1, qnrA, sul1, aaC, blaTEM), are still necessary.”
In addition, other concerns, as study limitations, has been highlighted in the revised version of the manuscript: “Even if the presented preliminary results of the present study can be considered suggetive, further investigations, focusing on the molecular evidence of other enterohemorrhagic (e.g. eaeA), enterotoxigenic (e.g. elt, est), enteroinvazive (e.g. invE), enteroaggregative (e.g. Eagg, astA), and diffusely adherent (e.g. daaD) virulence factors/genes are still required for a better understanding of the pathogenic potential of the raw milk cheese origin E. coli strains.”
Out of these strains, in the present study and within the available financial resources, the authors focused on the detection of STEC/VTEC strains due to their toxicity and infectious power, able to generate serious infections in humans. Nevertheless, the fact that Romania regularly reports the occurrence of VTEC/STEC infections in humans within the European Centre for Disease Prevention and Control (ECDC) surveillance system opens the opportunity to monitor this pathogen in raw food sources, like cheeses.
(4) Line 74: What was the novelty of this work?
Answer: Until now, according to the authors knowledge, only one study was conducted in Romania in order to monitor the virulence potential and genotypic antimicrobial resistance of cheese origin E. coli strains in another region of the country. The present investigation is the first reporting phenotypic antimicrobial resistance profile of cheese origin E. coli strains towards a wide range of antimicrobials used in human and veterinary medicine. This fact was mentioned in the revised version of the manuscript “Presently, no available scientific report in our country about the phenotypic resistance pattern of the cheese origin E. coli strains”.
(5) Lines 88-89: What was the minimum number of pathogenic Escherichia coli to cause infections?
Answer: As response to the reviewer question, the following sentence was inserted in the revised version of the manuscript “It is important to mention that E. coli infection in humans requires low infectious doses, estimated < 100 cells.”
(6) Line 125: Was the antibiotic resistance of these isolates more serious than that in similar studies?
Answer: In terms of multi-drug resistance, the recorded MDR rate was higher in Iran (30.2%, [10]) and Saudi Arabia (100%, [28]), but lower in Northern China (19.4%, [26]).
Thank you again!
Reviewer 4 Report
This study aimed to investigate the occurrence, pathogenic potential, and antimicrobial susceptibility profile of E. coli isolates in raw milk cheese produced and commercialized in Banat region, Romani
- Line 47: reviewed by Kaper et al (3).
- The number of collected milk samples is low.
- How many cheese samples were collected.
- Provide the concentration of each antibiotic in 4.4 section.
- in section 4.4; group the antibiotics based on the class.
- Add the breakpoint for each antimicrobial in supplementary materials.
- You should do genotypic analysis for the resistance genes contained in these samples and then make a correlation between genotypic and phenotypic analysis.
- Provide more details on the biochemical test for isolates confirmation.
- Table 2 is confusing, please divide it into two tables.
- Why did you use two different AST cards?
Author Response
Dear Editor and Reviewer,
We would like to thank the Editor and reviewer for their time, interest and comments to our manuscript. We are delighted to be considered for publication in the prestigious Antibiotics-Basel journal and to receive quality peer reviews that have greatly improved our manuscript. We have thoroughly considered each Reviewer's comments and have revised the text to reflect this. We have addressed each Reviewer’s comments below, in a point-by-point fashion. The new version of the manuscript contains all of the requested and operated changes and additional information (as reviewers required) highlighted with red font. Also, following the recommendations of a reviewer, the manuscript was checked by a native English-speaking colleague.
Below, you can find the answers to the all comments and suggestions.
Thank you again for your time and consideration.
Sincerely yours,
Dr. Adriana MORAR
On behalf of all the authors
Reviewer #4
This study aimed to investigate the occurrence, pathogenic potential, and antimicrobial susceptibility profile of E. coli isolates in raw milk cheese produced and commercialized in Banat region, Romania
Answer: Thank you very much for your overall positive appreciations and efforts to review our manuscript!
-Line 47: reviewed by Kaper et al (3).
Answer: was done, as requested!
-The number of collected milk samples is low. How many cheese samples were collected.
Answer: The research team members completely agree the reviewer opinion that, overall, the number of collected and processed cheese samples is low. This concern, is related by the existent very limited financial resources. However, the total number of collected samples (n=53) were provided by a representative number of cheese producers, offering a relatively complex image in term of territorial distribution about the hygienic – sanitary quality of the cheese assortments which they produce. Also, this fact can result in a more complex understanding of the antimicrobial resistance phenomenon of the isolated E. coli strains, originating from cheeses produced by different farmhouse cheesemakers (the herds of different cheese producers receive different antimicrobial treatment protocols). Being a preliminary investigation, the sampling strategy tried to focused on the increasing of the sample origin diversity, and not on to collect a high number of samples in different periods within a large interval from a low number of producers.
- Provide the concentration of each antibiotic in 4.4 section.
Answer: According to the reviewer requirement, the concentration for all of the tested antimicrobials was inserted!
-in section 4.4; group the antibiotics based on the class.
Answer: According to the reviewer requirement, the antibiotics were grouped/presented based on the classes they belong to.
-Add the breakpoint for each antimicrobial in supplementary materials.
Answer: According to the reviewer requirement, the breakpoints for each antimicrobial was added in supplementary materials.
-You should do genotypic analysis for the resistance genes contained in these samples and then make a correlation between genotypic and phenotypic analysis.
Answer: With respect to the reviewer request, the research team, presently don’t have the financial support to fulfill this request. However, this concern, as study limitation, has been highlighted in the original submitted version of the manuscript (see the lines 208-211). In the revised version the sentence was slightly rephrased resulting in: „However, for a better understanding of the AMR phenomenon of this food-borne pathogen in our country, further studies, based on the monitoring and characterization of genotypic resistance pattern of the isolates, by highlighting the presence of antimicrobial resistance genes (e.g. aadA1, tetA, tetB, dfrA1, sul1, qnrA, aaC, blaTEM), are still necessary”
-Provide more details on the biochemical test for isolates confirmation.
Answer: In agreement with the reviewer requirement, the following sentences were inserted: „Species identification was done by testing the complete biochemical properties of the isolates with the Vitek2 (bioMérieux, Marcy l’Etoile, France) automated compact system, along with the reference strain E. coli ATCC 25922 as control, and using the Vitek 2® ID-GN (Gram-negative, ID Card No. 21311) specific identification cards, following the manufacturer instructions. This card include a total of 47 biochemical tests, beside one negative control well, measuring the carbon source utilization, enzymatic activities and resistance of the E. coli isolates.”
-Table 2 is confusing, please divide it into two tables.
Answer: According to the reviewer requirement the Table 2 was divided resulting in Tables 2 and 3 in the new version of the manuscript.
-Why did you use two different AST cards?
Answer: Thank you very much for this question! As the authors highlighted within the lines 281-283 of the original submission, the most of the isolates (n=28) has been tested with the card AST-N204, designed for antimicrobials used in human medicine. However, in order to (i) increase the study value, (ii) to widen the used antimicrobials diversity and (iii), also, to provide feedback to veterinarians, the authors decided that a smaller part of the isolates (n=15) to be tested with a Gram-negative card (AST-GN96) recommended for veterinary use only. The authors acknowledged that it would have been better that all isolates had been tested with both cards, but the existent financial resources were very limited. Precisely for this reason, the study was presented under the form of Communication, presenting preliminary, but significant results. The authors intend to continue the study. According to the reviewer request, the authors included a justification for using different Vitek cards, for testing antimicrobial susceptibility profile of the isolates, namely:
“Within this study, in context of the existence of very limited financial resources, the research team tried to expand the diversity of the used antimicrobials, in order to provide feedback to both physicians, as well as veterinarians on the resistance of the isolated strains towards routinely used antimicrobials. In this regard, two different Gram-negative specific cards, namely AST-N204 (for human use) and AST-GN96 (for veterinary use), were applied for the monitoring of antimicrobial resistance profile of 28 and 15 isolates, respectively”
Thank you again for your time and efforts!
Round 2
Reviewer 2 Report
The manuscript has been improved.
However, the discussion remains longer than necessary and verbose in comparison to the limited study. Before final acceptance, the discussion should be shortened significantly.
Author Response
We would like to thanks for their time, interest and comments to our manuscript. We are delighted to be considered for publication in the prestigious Antibiotics-Basel journal. We have addressed to reviewer comment in a point-by-point fashion. The new version of the manuscript contains all of the requested changes. Also, following the recommendations of another reviewer, the manuscript was checked by a native English-speaking colleague.
Below, you can find the answer to the reviewer comment.
Thank you again for your time and consideration.
Sincerely yours,
Dr. Adriana MORAR
On behalf of all the authors
Reviewer #2
However, the discussion remains longer than necessary and verbose in comparison to the limited study. Before final acceptance, the discussion should be shortened significantly.
Answer: Dear Reviewer, according to your request, the discussion chapter of the manuscript has been significantly shortened. All modifications are highlighted with blue font in the revised version of the manuscript.
Thank you again for your time and efforts!
Reviewer 4 Report
Thank you. Please make English proof
Author Response
Reviewer #4
Thank you. Please make English proof
Answer: Dear reviewer, according to your request, the manuscript was checked by a native English-speaking colleague. The new version of the manuscript contain all of the operated changes highlighted with red font.
Thank you again for your time and efforts!